# GSAP-NER: A Novel Task, Corpus, and Baseline for Scholarly Entity Extraction Focused on Machine Learning Models and Datasets

Wolfgang Otto[1], Matthäus Zloch[1,2], Lu Gan[1,2], Saurav Karmakar[1], and Stefan Dietze[1,2]

[1] GESIS – Leibniz Institute for the Social Sciences, Cologne, Germany
[2] Heinrich-Heine-University Düsseldorf, Germany
{wolfgang.otto, lu.gan, saurav.karmakar, stefan.dietze}@gesis.org
{matthaeus.zloch, lu.gan, stefan.dietze}@hhu.de

## Abstract

Named Entity Recognition (NER) models play a crucial role in various NLP tasks, including information extraction (IE) and text understanding. In academic writing, references to machine learning models and datasets are fundamental components of various computer science publications and necessitate accurate models for identification. Despite the advancements in NER, existing ground truth datasets do not treat fine-grained types like *ML model* and *model architecture* as separate entity types, and consequently, baseline models cannot recognize them as such. In this paper, we release a corpus of 100 manually annotated full-text scientific publications and a first baseline model for 10 entity types centered around ML models and datasets. In order to provide a nuanced understanding of how ML models and datasets are mentioned and utilized, our dataset also contains annotations for informal mentions like "our BERT-based model" or "an image CNN". You can find the ground truth dataset and code to replicate model training at https://data.gesis.org/gsap/gsap-ner.

## 1 Introduction

Throughout various disciplines, the scientific process constantly produces new knowledge, innovative discoveries, and valuable insights, which typically are published in conference proceedings and journal articles. The increasing volume of scholarly artifacts underscores the importance for scientists to efficiently locate, comprehend, and utilize these resources in their daily work. Consequently, the NLP community is constantly creating methods to extract named entities from the scholarly domain, recognizing its significance in facilitating scientific understanding.

With the advent of artificial intelligence (AI), the landscape of machine learning (ML) approaches has evolved, incorporating techniques such as deep learning (DL) and fine-tuning of large language

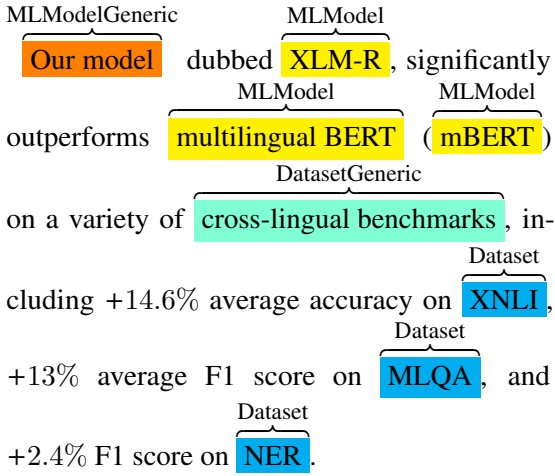

Figure 1: Example of machine learning models and dataset related mentions annotated according to our tag set. "Our model" and "cross-lingual benchmarks" are considered *informal mentions*, whereas the others are noun phrases mentioning named entities.

models (LLM). Consequently, to effectively comprehend scientific articles related to ML, AI, or data science, it becomes crucial to identify and comprehend these emerging entity types.

Developing effective NER models for these entities requires annotation guidelines and ground truth datasets to train robust language models (Qasem-iZadeh and Schumann, 2016; Luan et al., 2018). However, existing guidelines and ground truth datasets for scholarly entities have not adequately addressed the finer-grained entity types, such as *ML Models* and *Datasets* as distinct entities. Instead, state-of-the-art works treat ML models as *Methods* (Färber et al., 2021), failing to differentiate between the model instance, type, and underlying architecture. Similarly, dataset mentions are typically categorized as *Material*, overlooking the fact that this can also encompass knowledge bases, resources, or other corpora.

This paper presents GSAP-NER[1], a ground truth

---

[1] Acronym stands for: GESIS Scholarly Annotation Project

dataset specifically designed to enable the development of language models tailored for identifying named entities associated with the interplay between machine learning models and datasets. It benefits from a detailed annotation scheme that is customized for the discussion and use of machine learning models and the data used. We address the limitation of existing datasets by emphasizing comprehensive annotation of full scientific paper annotation rather than solely focusing on annotated abstracts or pre-selected sections. Our dataset offers two significant advantages. Firstly, we place particular emphasis on capturing *informal mentions* of named entities. These unnamed, descriptive mentions indirectly relate to named entities (e.g., "cross-lingual benchmarks" in Figure 1), providing valuable training data for co-reference resolution tasks. Secondly, our dataset features nested entity annotations (Finkel and Manning, 2009; Katiyar and Cardie, 2018), enabling the annotation of multiple sub-parts of a text span within a single noun phrase.

Based on our ground truth we fine-tune a first baseline model for our task of ML model and dataset named entity recognition. We employ three state-of-the-art baseline models for that, SciBERT (Beltagy et al., 2019), RoBERTa (Liu et al., 2019), and a more recent pre-trained language model called SciDeBERTa-CS (Jeong and Kim, 2022). We have found that SciDeBERTa-CS performs best on the entity types MLModel and Dataset, with an F1 score of 0.71 and 0.81, respectively.

Creating a sizable ground truth dataset like GSAP-NER is costly in terms of effort. Therefore, as a final experiment, we explore the minimum quantity of fully-annotated texts required to observe a noteworthy improvement in performance by incrementally increasing the size of the training data. Our dataset enables researchers and practitioners to extract precise and domain-specific information, contributing to fields such as information retrieval, scientific knowledge mining, automated literature analysis, and knowledge graph creation.

Our research presents four key contributions, each aimed at advancing the state of scholarly entity and concept detection:

- We provide a manually annotated dataset containing 100 full-text computer science publications with over 54k entity mentions in 25,857 sentences (Section 3).

|  | SciERC | SciREX | TDMSci | Heddes | GSAP NER |
|---|---|---|---|---|---|
| Ann.unit | ♣ | ♥ | ♠ | ♠ | ♥ |
| # pub. | 500 | 438 | n/a | n/a | 100 |
| # pos. sent. | 2,558 | 76,223 | 2,000 | 2,911 | 25,857 |
| # neg. sent. | 214 | 44,209 | 0 | 3,089 | 7,530 |
| # ent.types | 6 | 4 | 3 | 1 | 10 |
| # mentions | 8,089 | 156,931 | 2,937 | 3,729 | 54,598 |
| # ment./pub. | 16.2 | 358.3 | n/a | n/a | 546.0 |
| Material | ■ | □ | □ | □ | ■ |
| Dataset | □ | ■ | ■ | ■ | ■ |
| DataSource | □ | □ | □ | □ | ■ |
| Metric | ■ | ■ | ■ | □ | □ |
| Method | ■ | ■ | □ | □ | ■ |
| ML Model | □ | □ | □ | □ | ■ |
| ModelArch. | □ | □ | □ | □ | ■ |
| Task | ■ | ■ | ■ | □ | ■ |

Table 1: Comparison of ground truth datasets for scholarly NER Tasks including annotated entity types. Annotation units: ♣=abstract, ♠=sentence, ♥=full-text.

- We introduce a fine-grained tag set designed for detecting scholarly entities and concepts, customized to reflect the use and presentation of machine learning models and datasets in scientific publications (Section 3.1).
- We conduct a comprehensive performance evaluation of baseline models for our ten defined entity types (Section 4 and 5).
- We explore the minimum number of annotated publications needed to achieve satisfactory performance in our fine-grained scholarly NER task, which can guide future annotation projects (Section 6.2).

All materials, such as the ground truth dataset and the code to replicate model training can be found at https://data.gesis.org/gsap/gsap-ner.

## 2 Related Work

Among the works dealing with the task of scholarly information extraction, in this section we focus on those named entity recognition[2] methods which are machine-learning-based and not rule-based approaches. As a general overview, Nasar et al. (2018) gives a comprehensive list of approaches on information extraction from scientific publications.

Multiple datasets serve as ground truth datasets for Named Entity Recognition (NER), each cater-

---

[2]In the literature, it is also referred to as key-insight extraction, (typed) entity recognition, entity extraction, or (scientific) concept extraction.

ing to specific tasks. Please consult Table 1 for a comparison of the most related ground truth datasets to ours, GSAP-NER.

Among datasets working on abstracts, SciERC stands out as a prominent dataset, comprising 500 abstracts extracted from 12 AI conference and workshop proceedings (Luan et al., 2018). This rich dataset contains annotations for scientific entities, their relationships, and co-reference clusters, which are invaluable for related NLP tasks.

Another dataset, SciREX, offers comprehensive coverage with 438 fully annotated documents, specifically targeting mention identification and relationship extraction of entities related to methods, tasks, datasets, and metrics (Jain et al., 2020). To prepare their ground truth dataset for the NER task, they combined distant supervision and manual correction of automatically pre-annotated full-texts. In contrast to their work, we created a fully manually annotated corpus. This enables us to define our tag set independently of current approaches and to avoid potential model bias introduced by pre-annotation.

Hou et al. (2021) contributed significantly to this area by presenting TDMSci, a corpus containing domain expert annotations for TDM entities in 2000 sentences extracted from NLP papers, alongside a dedicated TDM tagger designed for this specific task.

Heddes et al. (2021) developed a ground truth dataset for dataset mention detection, comprising 6,000 annotated sentences selected by the occurrence of dataset related word patterns that were sourced from four major AI conference publications. Approximately half of them containing one or more named datasets.

An emerging NLP task known as leaderboard extraction focuses on extracting Task-Dataset-Metric-Score (TDMS) tuples from scholarly papers, enabling the generation of an aggregated comparison view of the main entities of interest (Kabongo et al., 2021). Along this direction, Kardas et al. (2020) introduced an extraction pipeline, AxCell, for extracting results from tables listed in scientific articles. In 2022, D'Souza and Auer (2022) created CS-NER, a corpus of contribution-centric information extraction targets, namely research problem, method, solution, dataset, metric, and more.

Recent lines of research have explored end-to-end frameworks based on NLP extraction tasks, such as NER, which involve a series of interconnected methods aimed at creating knowledge bases or knowledge graphs. Agrawal et al. (2019) focused on extracting the aim, method, and result sections from scientific articles, utilizing this information to construct a scientific knowledge graph. Similarly, Mondal et al. (2021) developed *SciNLP-KG*, a framework designed to extract TDM entities and relations from papers in the NLP domain. Furthermore, Dessí et al. (2022) presented a computer science knowledge graph (CS-KG) that is automatically generated and periodically updated. They achieved this by applying an information extraction pipeline to a vast repository of research papers, offering a comprehensive and up-to-date resource for the computer science domain.

## 3 Dataset

Machine learning models and datasets are essential scholarly entities that are discussed in various scientific disciplines. For named entities, it is frequently observed that, depending on context, identical string spans refer to different semantics. Take "BERT" in natural language processing as an example: it can refer to a particular pre-trained model with fixed parameters like "BERT Base" or its architecture, depending on the context. In addition to named entities, unnamed or informal mentions of machine learning models or datasets are more common in scientific text. But those informal mentions often contain nested references to other named entities and thereby carry extra information linking not only to machine learning models or datasets but also to other scholarly entities such as methods, model architectures and tasks. An illustrative example can be found in Table 2: "For the ResNets we train [. . . ]". The additionally carried information via informal, generic mentions requests a nested annotation style, meaning both the informal mention and the referenced nested named entity needs to be annotated. Therefore, in order to construct a gold standard scholarly entity mentions dataset, we have defined 10 different entity types in 3 categories: MLModel related, Dataset related and miscellaneous. The following gives a brief description of the entity types in our tag set. The more detailed annotation guideline includes further description, examples and figures, and can be found on the projects Web page[3].

---

[3] https://data.gesis.org/gsap/gsap-ner

| Annotation sentence with identified spans | Justification |
|---|---|
| For `the` `ResNets` we train a `ResNet-50`, a `ResNet-101`, and then `3 more` ... | `the` `ResNets` is annotated as a whole as informal mention of multiple concrete *MLModel*s. Additionally, it includes the nested annotation `ResNets`, which corresponds to the structural type information *ModelArchitecture*. `ResNet-50` and `ResNet-101` are actual executable *MLModel*s. |
| We publicly release `a new large-scale dataset`, called `SearchQA` ... (an existing question-answer pair is) crawled from `J!Archive`, and augment it with the text snippets retrieved by `Google`. | *Dataset* and *DatasetGeneric* are identified with similar reason to the previous example, while unstable data source information are recognized as *DataSource*. |

Table 2: Annotation examples for machine learning model related entities: `MLModel`, `MLModelGeneric`, `ModelArchitecture`. And data related entities: `Dataset`, `DatasetGeneric`, and `DataSource`.

## 3.1 Annotation Tag Set

We categorize our annotation tag sets into three categories: (1) MLModel related, including *MLModel*, *ModelArchitecture*, *MLModelGeneric*, *Method* and *Task*; (2) dataset related, including *Dataset*, *DatasetGeneric* and *DataSource*; (3) miscellaneous, including *ReferenceLink* and *URL*. In particular, the *Generic*s (*MLModelGeneric*, *DatasetGeneric*) correspond to the informal mentions of named entities.

### 3.1.1 MLModel Related

Machine learning model-related entities are tagged with this category of tags. We specifically separate ML pre-trained models from ML concepts, and map them into *MLModel* and *ModelArchitecture*. We explain each single of the entity tags below and illustrate them with a real world example from our annotation work in Table 2.

**MLModel** refers to a string span that represents a named entity of a machine learning model. For neural network based machine learning models, such a string span should correspond to an executable resource of the model in the context. In the first example of Table 2, "ResNet-50" corresponds to a trained executable resource and is therefore annotated as MLModel. A MLModel usually is based on some machine learning (ML) architecture, and can be applied to some ML tasks.

**ModelArchitecture** refers to a named entity corresponding to the conceptual or structural information of a machine learning model. ModelArchitecture can usually be interpreted as type information of other MLModel entities[4]. In the nested annota-

tion in Table 2, "ResNets" is labeled as a ModelArchitecture due to its abstract and categorical nature, rather than denoting a specific resource.

**MLModelGeneric** corresponds to the informal or unnamed mentions of MLModel entities. These informal mentions use possessive, temporal, quantitative or qualitative features refer to one, multiple or general MLModel entities.

**Method** corresponds to a non-MLModel approach, or a scholarly entity produced by MLModel entities and non-MLModel approaches (e.g., "word embedding"). This definition is in accordance with other annotation guidelines (SciIE, SciERC), which also define "Method" as a broad category of various methodological statements.

**Task** refers to a named entity of a machine learning task. We note that a task can relate to both ML models and datasets; a MLModel can be applied on a Task and a Task can be based on a Dataset. For simplification, we assign it under MLModel related category.

### 3.1.2 Dataset Related

Dataset related entities are tagged with this category of tags. We explain the entity tags below and illustrate their usage with real examples from our annotation work in Table 2.

**Dataset** refers to a named string span corresponding to an explicit dataset object in the text (e.g., "Social Bias Inference Corpus", "SBIC", "SQuAD").
**DataSource** corresponds a named entity of some unstable or unstatic data source information (e.g.,

---

[4]We differ a MLModel from a ModelArchitecture for a name entity essentially by whether the name entity is served as a resource/artifact or a concept/idea in the context. We

particularly assign a corresponding name entity as MLModel when it is mentioned for performance comparison, as shown in Figure 1. During the annotation process, we collect and categorize confusing and borderline cases according to the mention patterns, which we give a more detailed demonstration in the additional material.

"Google", "Twitter"). A DataSource is unstable due to its time-evolving nature and intractable timestamp. Therefore, knowledge bases and general mention of Wikipedia are considered as Data-Source. On the contrary, a Wiki dump with a specific timestamp will be annotated as Dataset.

**DatasetGeneric** corresponds to the informal or anonymous mentions of Dataset entities. Similar to MLModelGeneric for MLModel, DatasetGeneric entities use possessive, temporal, quantitative or qualitative features to refer one, multiple or general Dataset entities.

### 3.1.3 Miscellaneous

*URL* corresponds to a string span that is an URL in the text. *ReferenceLink* a string span that represents a reference in the text. A ReferenceLink may present in different style, but it requires to be linkable to the bibliography section at the end of the scientific article.

### 3.2 Publication Sampling

Selecting relevant and representative publications for the purpose of training and evaluating, which either introduce or harness machine learning models and datasets presents a dual challenge. On one hand, it necessitates the inclusion of cutting-edge methodologies and well-established models and datasets to ensure a comprehensive overview. On the other hand, it's equally crucial to embrace diversity by incorporating publications that might be less recognized or have garnered fewer citations, thus providing a broader spectrum of perspectives. For our primary source of full-text materials, we place our trust in arXiv[5], the preeminent open-access repository within the domain of computer science. In our quest to curate a selection of 100 publications, we employ two distinct but intertwined strategies: one that prioritizes popularity (1), and another that promote diversity (2).

Due to its popularity (1), we turn to Huggingface[6], a premier and dominant platform today for showcasing and distributing machine learning models (Jiang et al., 2023). Using the number of downloads as a metric, we compile a list of the most frequently used models. We then search the models' README files for links to publications on

| | mutual F1 exact-match | mutual F1 partial-match |
|---|---|---|
| MLModel | 72.1 | 74.6 |
| MLModelGeneric | 60.7 | 67.6 |
| ModelArchitecture | 23.7 | 34.4 |
| Method | 47.0 | 60.7 |
| Task | 51.4 | 55.2 |
| Dataset | 84.1 | 86.7 |
| DatasetGeneric | 56.2 | 65.8 |
| DataSource | 55.3 | 62.7 |
| ReferenceLink | 90.5 | 94.8 |
| URL | 86.1 | 94.1 |
| all | 61.4 | 69.3 |

Table 3: Interrator agreement as measured by the average mutual F1 of three annotators on the 14% co-annotated publications.

arXiv, including citation hints. This process results in a selection of 50 publications that not only present models available in Huggingface, but also discuss datasets, tasks, architectures and methods used.

To account for diversity (2), we randomly select out of model-related arXiv publications. To identify those, we filter the arXiv publications by research area (i.e., "cs.LG: Machine Learning"), based on keyword match and by time frame (i.e., first upload between 2018 and 2022).[7] For the keyword-based relevance classification, a heuristic is utilized. Publications must mention the term *model* in the title or at the beginning of the abstract (first 20 tokens), and *data* must be mentioned in the abstract.[8] Finally, we randomly selected 50 publications from the resulting pool of 12,641 arXiv publications. The final collection of publications is subjected to a validation process to ensure that it is not part of other datasets such as SciERC or SciREX.

### 3.3 Annotation Strategy

We have three annotators with computer science background to conduct the annotation using INCEpTION[9]. All three annotators had annota-

---

[5] https://arxiv.org

[6] We used Huggingface data as of November 13, 2022, as the basis for our selection. It is available at https://huggingface.co/models?sort=downloads.

[7] We used Version 123 of the arXiv Dataset available at https://www.kaggle.com/datasets/Cornell-University/arxiv/versions/123.

[8] Data is considered to be mentioned if one of the following terms is present in the abstract: "dataset", "datasets", "data", "corpus", "corpora".

[9] https://inception-project.github.io/

| | # spans | # unique spans |
|---|---|---|
| Method | 12,826 | 6,547 |
| DatasetGeneric | 9,838 | 5,781 |
| MLModelGeneric | 8,521 | 4,238 |
| ReferenceLink | 7,172 | 2,257 |
| MLModel | 5,012 | 944 |
| Task | 4,143 | 1,478 |
| Dataset | 3,898 | 883 |
| ModelArchitecture | 2,612 | 985 |
| DataSource | 508 | 185 |
| URL | 68 | 61 |
| Total | 54,598 | 23,359 |

Table 4: Text span statistics in our GSAP-NER dataset ordered by the number of spans per entity type.

tion training before starting to annotate on target publications. We randomly select $14\%$ of the publications for joint annotation by all three annotators. The rest of the publications are split and assigned to a single annotator each. The annotators identify mentions according to our tag set definitions, and nested annotations are allowed. For particular linguistic cases, we combine the reuse of ACL RD-TEC Guideline[10] and creation of new rules to adapt our annotation schema. The linguistic cases include but are not limited to articles, abbreviations, adjective modifiers, conjunctions and prepositions, and plurals. For articles like "a" or "the", annotators are instructed not to include them, except for generic mentions. Abbreviations and adjective modifiers, conjunctions and prepositions are generally requested to be annotated following the ACL RD-TEC Guideline. Most plural forms are considered to be of generic type, unless it is a named entity.

### 3.4 Interrater Agreement

We calculate interrater agreement to measure the annotation coherence of the 14 common annotated publications. For this, we report the average mutual F1 score. To compute this metric, we compare the annotations for each pair of annotators using the F1 score, where one annotator is the ground truth and the other is the prediction, and then reverse their roles. The F1 score is reported for exact and partial matches. Compared to "exact match", the "partial match" setting considers partially overlapping

[10]ACL RD-TEC Guideline:
https://doi.org/10.13140/RG.2.1.1939.1446.

spans as matches, enabling us to better comprehend the disparities in annotations as partial-match disregards dissimilar annotation boundaries as errors. Our exact match and partial match agreement scores for each entity type are presented in Table 3.

### 3.5 Corpus Statistics

Table 4 lists some statistics of our corpus annotations on 100 publications. We report the number of spans per entity type as well as the corresponding unique number of spans throughout the documents. In total, GSAP-NER contains 54,598 annotated spans out of which 23,359 are unique spans.

## 4 Baseline Model

### 4.1 Problem Definition

Our goal is to identify named entities in the full text of scientific publications. We denote the tag set as $T$, where $t_i \in T (i = 10)$ is a tag described in Section 3. For each publication $D$ the goal is to generate a list of $j$ entity mentions, identified by a tuple $m_j = (t_i, b_j, e_j)$, where $m_j$ represents the mention span, $t_i$ the type of the named entity, $b_j$ and $e_j$ the start and end of each span in $D$. Note that since nested annotations are allowed, two identified mentions $m_k, m_l, k \neq l$, can have overlapping spans. This problem definition is flexible enough to use both transformer-based architectures and generative approaches to NER.

### 4.2 Pre-trained Model Selection

NER approaches currently follow one of two competing NLP paradigms (Liu et al., 2023). The state-of-the-art models for NER in the scientific domain follow the "pre-train, fine-tune, predict"-paradigm (Jeong and Kim, 2022). This paradigm involves pre-training with out-of-task goals, such as masked token prediction (MTP), in an unsupervised manner and fine-tuning these pre-trained language models (PLMs) on the downstream NER task. In contrast, recent, popular in-context learning based approaches utilize the "pre-train, prompt, predict"-paradigm and generative large language models are proposed for NER. However, Ye et al. (2022) show that they are not yet competitive for domain-specific downstream extraction tasks, such as NER on scholarly documents. Therefore, we present a baseline comparison based on state-of-the-art fine-tuning approaches of PLMs, which are proven to outperform previous approaches (Heddes et al., 2021; Jeong and Kim, 2022) in this domain.

To set a strong baseline that accompanies our GSAP-NER corpus, we conducted a comparative analysis of three baseline models. The initial model was chosen as a benchmark, reflecting a well-established foundation within the field. In particular, SciBERT (Beltagy et al., 2019) has consistently demonstrated good performance in various scholarly NER tasks, rendering it the default model for pre-training in this specific domain. It is a version of BERT (Devlin et al., 2019) additionally pre-trained on scholarly documents using a Multi-Task Prediction (MTP) objective and has a parameter count of 109 million. Subsequently, we fine-tuned DeBERTa-CS (Jeong and Kim, 2022), a more recent iteration of an in-domain pre-trained Language Model. This choice was motivated by the authors' track record of achieving state-of-the-art results in NER tasks on datasets such as SciERC (Luan et al., 2018). Like SciBERT, SciDeBERTa-CS was also pre-trained within the scientific domain, with a more focused emphasis on the Computer Science (CS) domain. This model employs a configuration consisting of 125 million parameters. To enable a comparative performance evaluation with pre-trained foundation models not specialized for the scientific domain, we fine-tuned two RoBERTa model versions, "Base" and "Large", which comprise 125 million and 355 million parameters, respectively. These models leveraged a dynamic masking strategy and were pre-trained on a larger training corpus (Liu et al., 2019).

## 5 Experimental Setup

### 5.1 Preprocessing

For our experiments, we choose paragraphs as the processing unit instead of sentences, providing the models with more contextual information. However, the used PDF-to-text tool[11] introduced some errors for footnotes, figures or tables, during conversion to text. Therefore we asked the annotators to mark these errors as corrupt. Consequently, we exclude all paragraphs containing corrupt parts from train, validation and test set. The generated paragraphs exhibit an average length of approximately 4.5 sentences. The average token count per paragraph stands at 109.1, with a median of 88 tokens. Notably, only 16 paragraphs exceed the threshold of 512 tokens, as determined by tokenization using the SciBERT tokenizer. This enables their usability across various language models.

---

[11]grobid: `https://github.com/kermitt2/grobid`.

We transform the annotated spans into token wise labels based on the BIO tag scheme. As described in Section 3, our annotation guideline allows nested annotations, which presents challenges for out-of-the-box NER models. This is because the task of classifying each token becomes a multi-label classification problem rather than a multi-class classification problem. Analyzing on our GSAP-NER corpus showed two major patterns of nested annotations. The first involves named entities nested in *Generic*s, while the second relates to benchmark entities that are double annotated for both *Dataset* and *Task*. To address the first case, we split the entity tag set into two parts: *Generic* mentions and all other type of mentions. Additionally, we simplify the double annotated benchmark entities by converting them into a single *Dataset* span in the training set. This approach enables us to use two separate models for each of the tag set, resolving the nested entity annotation problem An analysis showed that the simplifications need to be done for generating the training data leads to a upper bound of 98.7% F1 score when testing the performance of the simplified tag set on the full annotations without any simplification. While finetunig PLMs we use a final fully connected layer on top of the encoding layer and trained to predict one label for each token using a cross entropy loss for each of the two models.

### 5.2 10-fold Cross Validation

The broader perspective of our model is to solve the NER task on whole documents, even if the unit or processing for our model is one paragraph. We consider 10-fold cross validation where folds are created such that all paragraphs from one publication are present in the same fold. In each cross validation round 80% of publications are used for training, 10% for validation and the remaining 10% for testing. For reasons of reproducibility, we publish the publications used in each fold as part of the data set.

### 5.3 Metrics

We evaluate our models with entity-level F1 score, where each entity annotation is identified by the *paragraph id*, *start index*, *end index*, and *label*. Gold annotations and predictions are represented as sets of entity annotations, and the F1 score is calculated based on these sets. For comparison, we also employ partial-match F1 score, which considers a predicted entity span as a match if it overlaps with

| | exact-match F1 | | | | partial-match F1 | | | |
|---|---|---|---|---|---|---|---|---|
| | SciBERT | SciDeBERTa-CS | RoBERTa-Base | RoBERTa-Large | SciBERT | SciDeBERTa-CS | RoBERTa-Base | RoBERTa-Large |
| MLModel | 60.8 | **70.1** | 67.1 | 69.3 | 63.5 | **73.0** | 70.1 | 71.7 |
| MLModelGeneric | 68.0 | **70.1** | 68.7 | 68.6 | 74.4 | **76.5** | 75.5 | 75.5 |
| ModelArchitecture | 30.9 | **33.9** | 30.6 | 30.2 | 44.7 | **48.3** | 45.6 | 44.9 |
| Method | 44.7 | **47.6** | 46.0 | 47.3 | 60.2 | **62.5** | 61.2 | 62.2 |
| Task | 52.1 | **55.3** | 52.8 | 53.7 | 59.3 | **60.8** | 59.5 | 60.5 |
| Dataset | 72.6 | **81.7** | 78.0 | 80.5 | 77.4 | **85.5** | 81.9 | 84.0 |
| DatasetGeneric | 63.3 | 63.2 | **63.8** | **63.8** | 73.4 | **73.6** | 73.5 | 74.2 |
| DataSource | 41.7 | **51.6** | 48.6 | 49.4 | 48.8 | **59.9** | 56.3 | 57.6 |
| ReferenceLink | **95.9** | 92.3 | 92.2 | 90.4 | **98.0** | **98.0** | 97.8 | **98.0** |
| URL | **68.3** | 50.5 | 64.9 | 32.8 | 85.0 | 64.1 | 77.2 | **85.2** |
| all | 61.9 | **64.6** | 63.0 | 63.5 | 70.6 | **73.4** | 72.0 | 72.7 |

Table 5: F1 performance comparison of four fine-tuned pre-trained language models (PLMs) for scholarly entity and concept detection. The metrics are calculated using 10-fold cross-validation. As performance measurements, we provide results for both exact matches and partial matches. The latter considers any overlap between the predicted outcome and the correct annotation as a match.

a gold annotation of the same label. This metric is particularly useful for assessing the correct position of entity annotations, even if the beginning and end of the tag span are not precisely predicted. To measure performance for each entity type separately, we only consider gold annotations and predictions for a specific label.

## 6 Experimental Results

### 6.1 Baseline Models

Fine-tuned PLMs proved to perform well on NER tasks for scholarly document processing. The F1 score performance overview of the four compared models (i.e., SciBERT, SciDeBERTa-CS, RoBERTa-Base, and RoBERTa-Large) show a general applicability on the given task. With exact-match F1 score in the range from 61.9 to 64.4 (Table 5), the performance is comparable with other annotation approaches (Luan et al., 2018). The best performing model SciDeBERTa-CS outperforms the much bigger RoBERTa model for nearly every entity type. Nonetheless, the performance varies in terms of entity types and models. To assess the significance of SciDeBERTa-CS model's

performance enhancements relative to the other models in our ten-fold cross-validation setup, we conducted a paired t-test. The obtained p-values for all comparisons were found to be below the significance threshold (commonly set at $0.05$), indicating a statistically significant difference in performance. To characterize the similar performance differences among all models for different labels, we identified two distinguishing criteria for entity types. Firstly, concrete named entities (e.g., MLModel: $70.1\%$ F1 and Dataset: $81.7\%$ F1) exhibit superior performance in contrast to conceptual entities (e.g., Method: $47.6\%$ F1 or ModelArchiteture: $33.9\%$ F1), irrespective of the quantity of training samples. Notably, the Method entity type is the most prevalent, encompassing over 12,500 annotated text spans (as shown in Table 4). The second criteria employs the presence of structural anchor points to distinguish between entity types. For instance, standardized patterns such as citations can be easily identified with a supervised approach. The existing weak URL extraction performance can be attributed to the limited number of annotations and the fragmented nature of URLs

generated during the conversion from PDF to text. Inaccuracies in rare URL annotations can significantly increase the error rate, whereas fragmented URLs pose challenges in accurately detecting the beginning and end of a URL. Another valuable insight arises from the observed variation in performance between exact-match and partial-match metrics. The performance gap ranges from $+2.9\%$ on MLModel to $+14.4\%$ on Method, emphasizing the significance of distinguishing between concrete and conceptual entity types. Recognizing the correct text spans for conceptual entity types remains a challenge for the models due to the uncertainties among the annotators. Please refer to Table 3 for comparison.

### 6.2 Train Size Experiment

In comprehensive full-text annotation initiatives, the quantity of annotated full-texts assumes a pivotal role owing to the substantial cost advantages linked with annotating a smaller subset of publications. To explore this facet, we executed an experiment aimed at assessing performance metrics across various training set dimensions. For the sake of efficiency, the SciBERT model was used in this experiment. In our 10-fold setup, we conducted fine-tuning for ten distinct models within each fold. In every iteration, a designated proportion of documents from the training dataset was allocated to individual models. The number of training documents varied in each iteration, ranging from 8 to 80, with increments of eight. The findings from this experiment are visually represented in Figure 2.

When assessing the F1 score, we observed a sub-

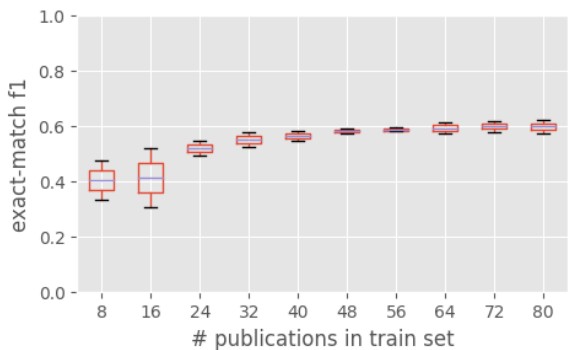

Figure 2: Increasing overall exact-match F1 performance of the SciBERT model trained on varying number of publications. The train set size in the 10-fold set up ranges from 8-80 publications in every fold. The box-plot illustrates the performance differences across folds.

stantial standard deviation in the case of models with a limited number of publications within the training dataset. Conversely, for models trained with more than 24 documents, resulting in a dataset of greater variability, this phenomenon was markedly mitigated. Subsequently, our investigation revealed that the fluctuations in the exact-match F1 score demonstrated a diminishing trend, stabilizing after the incorporation of more than 40 publications for training.

## 7 Conclusion

We introduce GSAP-NER, a manually annotated corpus over full-text scholarly publications from the computer science domain, designed for information extraction of ML models and datasets. By distinguishing ML models from methods and datasets from materials, our dataset enables researchers and services to gain deeper insights into the specific methods and materials employed in computer science research. We utilized our data and fine-tuned three state-of-the-art baseline models. The experiments showed that SciDeBERTa-CS reaches best performance on the majority of entities types, with an overall F1 score of $0.64$ and $0.73$ on exact span matches and partial span matches, respectively.

Despite the challenges involved in its creation, we believe GSAP-NER remains a valuable resource for the development, evaluation, and benchmarking of NER models in the computer science domain. It offers researchers and practitioners a comprehensive and domain-specific dataset, addressing the limitations of existing datasets that often lack specialized entity differentiation. Furthermore, this dataset can contribute to advancing research in areas such as information retrieval, scientific knowledge mining, automated literature analysis, and knowledge graph creation.

As future work, we aim to employ a multi-stage model training approach and leverage additional background knowledge to disambiguate syntactic mentions from their semantic context. Such background knowledge could come from ML ontologies and knowledge graphs, such as the ORKG or CS-KG. We also envision exploring entity relationships, co-reference resolution, and entity attributes as future directions to enhance the value of this dataset.

## Limitations

Despite our diligent efforts, developing a gold standard dataset for entity extraction using a fine-grained and comprehensive tag set focused on machine learning models and datasets remains a non-trivial undertaking. This leads to the following limitations associated with the creation of our corpus. First, our work suffers from low interrater agreement on certain entity types, and thus, the model performs poor on those types. For instance, when frequently used models such as "RoBERTa" are mentioned, it can be difficult to determine whether to classify them as ModelArchitecture or MLModel, depending strongly on the context. Efforts to address ambiguous types in the annotation guideline or increased training time of annotators did not solve this issue. Second, the paper selection is conducted within the machine learning domain and does not include infrequent publication types, such as surveys or reproducibility studies. Furthermore, the potential applicability of our approach across various research domains remains a topic for future investigation. Finally, during the model training process, we excluded paragraphs that were identified as erroneous by the annotators. It is essential to address and resolve the resulting issues before the model can be effectively used in a productive real-world setting.

## Ethics Statement

The authors foresee no ethical concerns with the work presented in this paper.

## Acknowledgements

We thank the anonymous reviewers for their constructive feedback and intense rebuttal phase. This work has been partially funded by the Deutsche Forschungsgemeinschaft (DFG, German Research Foundation) as part of the Projects BERD@NFDI (grant number 460037581), NFDI4DS (grant number 460234259), as well as Unknown Data (grant number 460676019).

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
