# OpenReview forum: "GSAP-NER: A Novel Task, Corpus, and Baseline for Scholarly Entity Extraction Focused on Machine Learning Models and Datasets"
_EMNLP/2023/Conference — EMNLP 2023 Findings_

### Official Review · Reviewer_CSpT · 2023-07-27

**Soundness:** 3

**Excitement:**

2: Mediocre: This paper makes marginal contributions (vs non-contemporaneous work), so I would rather not see it in the conference.

**Missing References:**

n/a

**Paper Topic And Main Contributions:**

The paper presents a newly constructed dataset for (named) entity recognition task specifically from scientific literature (focused on extracting models, dataset and coreferenced mentions thereof). This new dataset comprises 100 papers, that jointly comprise ~55k mentions, in ~25k sentences. They define their own entity types (adding new types 'Datasource' and distinguishing 'MLModel' and 'MLModelArchitecture', which were not represented in earlier similar efforts in literature).
The authors also provide baseline models for the NER task, based on existing pre-trained language models (both general purpose, Roberta, and specifically for scientific literature, SciBERT, and even more tailored to computer science, SciDeBERTa-CS).
The dataset is presented in terms of briefly stating the entity types and indicating human inter-annotator (dis)agreement.
Baseline model results on the defined NER task are presented.

**Questions For The Authors:**

- Can you comment on models exhibiting super-human performance? I.e., they beat inter-annotator agreement results.
- Can you provide a clear-cut definition of MLModel vs ModelArchitecture, which unambiguously allows annotators to differentiate them? (I found the annotation guidelines only partly satisfying in this respect)
- Can you please explain the model architecture, i.e., what you put on top of the BERT models, to finally output the NER spans?

**Reasons To Accept:**

- New dataset, with slightly finer granularity entity types compared to the most related (but 3x larger) SciREX dataset

**Reasons To Reject:**

- Motivation for the new dataset -- whose unique feature seems to be that it is specifically targeted at computer science, and particularly machine learning -- is only partly convincing; why would SciREX's 'method' not suffice in practice? Especially since human annotators seem to have difficulty agreeing on correctly labeling MLModel vs ModelArchitecture instances?
- The NER models are not explained in sufficient detail (and basically I'd be at loss when I'd try and reproduce the author's work based on the paper alone); the only thing that is clear is that they use BERT-based models; but how the BERT token representations eventually lead to (potentially overlapping) span annotations is not explained (not even with a reference to a paper that provides the model architecture)
- The definitions of the newly defined types, esp. MLModel vs ModelArchitecture are not clearly articulating how exactly they differ from one another (I also did not find a very unambiguous definition in the annotation guidelines); apparently, annotators face the same difficulty, given the low agreement
- The low inter-annotator agreement (on the newly defined entity types) is just stated as is, without any insight into whether related datasets (as referenced in the paper) have clearly higher/lower/comparable agreements.
- The fact that the baseline models outperform inter-annotator agreement results is not commented on (I'd expect these agreements to be regarded as upper bounds that a system could reach; wouldn't significantly higher model scores suggest over-fitting of some kind?)
- No numerical or qualitative information is stated on (statistical) significance of reported performance differences
- No clear discussion of the work's (potential) limitations (not for the dataset, nor for the baseline models)

**Reproducibility:**

3: Could reproduce the results with some difficulty. The settings of parameters are underspecified or subjectively determined; the training/evaluation data are not widely available.

**Reviewer Confidence:**

3: Pretty sure, but there's a chance I missed something. Although I have a good feel for this area in general, I did not carefully check the paper's details, e.g., the math, experimental design, or novelty.

**Typos Grammar Style And Presentation Improvements:**

- See my 'Questions for the authors'; such clarifications should be incorporated in the paper itself as well
- Please use curly quotes; and where you already do, please use the correct style for opening vs. closing ones (e.g., in the abstract)
- Please use a comma after "i.e." and "e.g."  (this is customary in US English, AFAIK)
- Please always put footnote superscript markers *after* punctuation (e.g., line 298)
- Table 1: please use the exact same terminology as where you define the precise entity types ("ML Arch." vs "ModelArchitecture")
- §1:
  + 'stacked annotations ...' -> maybe rephrase as 'mentions comprising overlapping spans' (this would be more clear to the general readership?)
  + 'very expensive': please quantify this, if possible (how many hours are we talking about? whether that's 'expensive' may be subjective)
  + l 110: tailored to around -> tailored to
  + Why are section references typeset in bold?
  + Fig 1 caption: rather than cite text parts in bold, just put them between quotes.
- §2:
  + footnote 1: may also refer -> also refers
  + 143: NLP-related tasks -> related NLP tasks
  + 159: more.. -> more.
  + it seems you sometimes typeset numbers in math mode, sometimes not. I'd simply never do it in plain text
- §3
  + 205: architecture depending -> architecture, depending   [add comma]
  + Table 2 caption: instead of typesetting "Pink spans" in pink, and later mention what it means (MLModel), just write the meaning formatted as in the table (i.e., a pink box around "MLModel" in this case); same for other color labels; would be even better to have them as a legend in / next to the table, rather than in the caption
  + ModelGeneric: "informal or anonymous": this is not clear; it seems you mean non-named (or under-specified) entity mentions that (co)refer to a specific named entity (ie. a MLModel, or ModelArchitecture?); I'd rather use 'ModelReference' as type name
  + Method:  also is not very clear; seems weird to combine (non-ML) models as well as their outputs (e.g., the final layer's token embeddings produced by BERT would also be a "method")?
  + Datasource:  "unstable or not static": not 100% clear what this means; what about Wikipedia dumps? The timestamped dump is static, right? But it would still be a web source, so 'datasource' and not 'dataset'?
  + DatasetGeneric: same comment as for ModelGeneric
  + On the 'popularity' selection: wouldn't this introduce a bias towards 'easy' (because frequently referenced, i.e., a model could basically memorize the names) instances?
  + On the paper selection strategy:  is the test set also stratified, i.e., the "popular" and "diverse" paper proportions therein are also 50/50?
  + 319: token -> tokens
  + 329: why exactly 14%? Seems like a random number
  + 338: not exhaustively -> are not limited to
  + 341: ... them. However, articles should be considered for generic mentions only. -> ... them, except for generic mentions.
  + 344: asked to annotate -> requested to be annotated
  + 358-61: The difference between exact match and partial match is, that partial match also consider the spans that are partial overlapping, which can evaluate more effectively on stacked annotations. --> Compared to ``exact match'', the ``partial match'' setting also considers partially overlapping spans as a match, which allows to more effectively evaluate stacked annotations.
  + 361: how do you define "more effectively"?
  + 362: And in -> In
  + 367: report on the -> report the
- §4
  + 382: are allowed to be -> can be
  + 385: "encourage": what do you mean exactly? How can a problem definition 'encourage' a model?
  + 399: "ready at dawn":  I don't know what you are trying to say
  + 400:  "the changing of the guard":  what do you mean exactly?
  + 402: tasks like -> tasks such as
  + 411: "three different aspects":  I'm not sure what you mean with 'aspects'; it seems that you refer to 3 different models (each with their own distinct characteristics), which you have chosen for a particular reason (i.e., there's an associated research question) --> rephrase: state the 3 research questions (or criteria for your model choices) + then list the corresponding models
  + 425: representing the aspect to compare a state-of-the-art model -> representing a state-of-the-art model [?]
- §5
  + 445: start of sentence seems to be missing?
  + 450: analysing -> analyzing  [in US English at least]
  + 462: show, that -> showed that
  + 464: S1 -> F1
  + 475: rest 10% -> remaining 10%
- §6
  + 501: "comparable with": please be specific; how much % does the difference amount to?  ('comparable' is subjective)
  + 513: sheer number -> number
  + 520 etc.: the explanation for low URL performance suggests it is a dataset issue (i.e., a result of poor PDF to text conversion)? A priori, I'd expect URLs to be very easy to extract, e.g., even just regex based. If it's a dataset issue, shouldn't we focus on correcting the data, rather than trying to have the model deal with it?
  + Table 5: Please report stdev (or some other indication that allows to assess the observed variation) over the 10 folds; this will also allow to comment on statistical significance of performance differences observed across different models.
  + Fig.2: What datapoints are the box plots based on? Are these different subsets? Are all boxplots taken over the same number of datasets?
  + 533: "the difference": unclear what difference exactly you're referring to here
  + 551: "a reasonable standard deviation": what is 'reasonable' exactly? Are you talking about the left bars in Fig.2? Or previous results (which did not include any stdev information)?
- §7
  + 572: the stated F1 values exceed those for inter-annotator agreement. Does that make sense?
  + 587: "multi-stage": please indicate what these 'stages' would be. Are you talking about a pipeline, e.g., first identifying spans, then classifying their types? Or rather also extract 'relations'? What would these be?
  + 589: "background knowledge":  this is very broad/vague; please be more specific

---

> ### Author Rebuttal · Authors · 2023-08-29
>
> We would like to thank you for your effort, your time, and your detailed, and constructive comments.
> Especially your detailed line by line comments are very helpful.
> We have gone through all the comments in detail and discussed all points among all co-authors. Below you will find our rebuttal for the issues raised by you.
> We comment on all your concerns on our paper based on your “Reasons to Reject” section. Doing so, we face all three questions in your “Questions to the authors” section. We point to the related questions in our comments.
>
> ## Resonse to reported key weaknesses:
> ### 1. Motivation - Missing novelty compared   to existing work, e.g. SciREX
> We decide to go for a new annotation corpus for full-text publications, to consider two major shortcomings of existing corpora like SciREX. (a) The lack of expressiveness towards named entity mentions of machine learning models and datasets. Our finegrained entities allow the possibility to detect mentions of MLModels and, for example, be able to capture relations between MLModel and Datasets (futurework). And (b) the shortcomings of an annotation pipeline using automated pre-annotated model suggestions and correct them by the annotator. We wanted to bring a novel source to the community.
> The fact that human annotators have issues on specific entity types is a sign, that the task is not easy, which leads to further research questions. But it should not lead to cancelation of further research on this topic. The comparable good performance on MLModel adds value to the research community on its own and enables better research for example on the interplay of MLModels and Datasets.
> ### 2. Missing Explanations of NER Models (Q3: Model Architecture not explained)
> We transform the annotated spans into token wise labels based on the BIO tag scheme. As in ll 461 described we split up the tag set and trained two expert models for each of the tag sets. We used a final fully connected layer on top of the encoding layer and trained to predict one label for each token using a cross entropy loss. To be able to reproduce our results we shared our code (link in abstract).
> We will incorporate a model architecture description of our NER approach into the final version of the paper.
> ### 3. Definition: MLModel vs. ModelArchitecture  (Q2: Clear Definition)
> We see a model as an executable piece of software that can be applied on a dataset . For instance, the mention "BERT-base" is a typical MLModel. On the other hand, ModelArchitecture is used for introduced terms or names that designate the architecture of a model. For architectures mentioned by name, there must additionally be a clear indication in the context that the concept of the architecture and its properties are explained and not a concrete instance of the model architecture.
> ### 4. No explanations or comparison of low inter-annotator agreements
> This is a good point to contextualize our inter-annotator agreements and further discuss the resulting limitations  which will be incorporated in the final submission.
> ### 5. Super human performance (Q1: super-human performance)
> First of all we need to differentiate the two values you are comparing. The first is based on annotations of our three annotators of 14% of the publications. The underlying data is different to the model scores calculated in the 10-fold setup.
> Also the metric is based on the mean mutual f1 scores between the annotations. Which have potential side effects like a huge influence of annotator outliers with huge differences in the annotations on the f1 means of specific entities.
> Anyway, to better be able to compare those numbers to explain this apparent super human performance we compared the model performance on the 14 commonly annotated publications on all three annotators annotations. We found that the models tends to find a common ground of the annotations for low entity types with low interrater agreement.
> ### 6. Missing report of statistical significance
> This is a valid point for comparing the different finetuned PLMs. Because of our 10-fold setup we decided not to have multiple training runs because of the immense number of models to train. For the final submission we will report and discuss the standard deviation of the performance between the folds. Even if this has the disadvantage that they are not directly comparable to performance standard deviations based on multiple runs on the same training data.
> ### 7. Limitations of dataset and baseline models
> We will incorporate a limitation section in the final version of the paper at least facing:
>  * the shortcomings of the used fine-grained schema
>  * limitations of the model concerning specific entity types
>
> ## Question to the authors:
> We have commented on all your Question above:
>  * Question 1: Point 5
>  * Question 2: Point 3
>  * Question 3: Point 2
>
> We have perceived your extensive comments which enable us to greatly improve the quality of our paper. We will go through them very conscientiously and incorporate your suggestions.

---

### Official Review · Reviewer_Z9Pn · 2023-08-04

**Soundness:** 4

**Excitement:**

3: Ambivalent: It has merits (e.g., it reports state-of-the-art results, the idea is nice), but there are key weaknesses (e.g., it describes incremental work), and it can significantly benefit from another round of revision. However, I won't object to accepting it if my co-reviewers champion it.

**Paper Topic And Main Contributions:**

This paper introduces a new dataset GSAP-NER covering unique machine learning-related entities such as Method (e.g., "word embedding"), MLModel (e.g., "XLM-R"), DatasetGeneric (e.g., "cross-lingual benchmarks"), Dataset (e.g., "MLQA"), and more. The dataset contains 54,598 entity mentions across 100 publications. Three annotators labeled the dataset 14% with good agreement, and the rest of the dataset was labeled independently. Finally, the paper introduced several baselines on the data that can be used by researchers looking to analyze mentions, or the baselines can be improved on by researchers looking to develop new methods.

**Questions For The Authors:**

Question A: What is the data card for this dataset?

**Reasons To Accept:**

Overall, the paper is well-written, and the data and annotation process/reasoning are well-explained. I can imagine many positive use cases for the dataset. For example, researchers could measure the popularity of methods in specific domains, indicating which domains certain approaches work best on. Moreover, the strong baselines the paper introduces will be useful for future researchers.

**Reasons To Reject:**

The paper is missing a complete "Data Card" for the new corpus [1]. This is vital for new datasets. When should the database used? What use cases should it not be used? What is the distribution with regard to the "perceived" author gender in the 100 papers? All this information is important.

(Minor) It would have been useful if a simple analysis was completed of how the dataset could be used for unique analysis. I think this would have been more useful than the model comparison. Overall, I think this dataset will have a large impact on practical analysis research. From a technical perspective, I don't think this dataset will add much to the existing NER datasets in terms of novel methodological development, unless the reasoning is missing (e.g., harder overlapping entites, non-contiguous entites, or something else).

[1] Pushkarna, Mahima, Andrew Zaldivar, and Oddur Kjartansson. "Data cards: Purposeful and transparent dataset documentation for responsible ai." Proceedings of the 2022 ACM Conference on Fairness, Accountability, and Transparency. 2022.



**Reproducibility:**

4: Could mostly reproduce the results, but there may be some variation because of sample variance or minor variations in their interpretation of the protocol or method.

**Reviewer Confidence:**

4: Quite sure. I tried to check the important points carefully. It's unlikely, though conceivable, that I missed something that should affect my ratings.

---

> ### Author Rebuttal · Authors · 2023-08-29
>
> Thank you for your valuable comments and your excellent suggestion. We agree that this dataset should have a data card too. We did not add it, as we do not have so much experience with writing data cards so far. We will create the data card for this dataset in case of acceptance.
>
> ### RE your question why we did not introduce a sample analysis illustrating how this dataset can be used?
> We plan an in-depth study applying our model on a bigger corpus of publications. This study is planned to cover 100k papers from various domains. This paper is just about the resource, i.e., the ground truth dataset itself, which we think is very valuable for the community, for example, as a second source of truth (next to other ground truth datasets like SciREX, e.g.).
> Thank you for your Review. We are looking forward to incorporating your valuable hints.

---

### Official Review · Reviewer_RRze · 2023-08-10

**Typos Grammar Style And Presentation Improvements:** I think it's mostly okay, noticed an …
**Soundness:** 4

**Excitement:**

2: Mediocre: This paper makes marginal contributions (vs non-contemporaneous work), so I would rather not see it in the conference.

**Missing References:**

For the stacked annotation part, I would say:
1. Finkel, J. R., & Manning, C. D. (2009, August). Nested named entity recognition. In Proceedings of the 2009 conference on empirical methods in natural language processing (pp. 141-150).

Also, possibly related to the general theme of the paper:

2. Kardas, M., Czapla, P., Stenetorp, P., Ruder, S., Riedel, S., Taylor, R., & Stojnic, R. (2020). Axcell: Automatic extraction of results from machine learning papers. arXiv preprint arXiv:2004.14356.

**Paper Topic And Main Contributions:**

The paper a) creates a new dataset by annotating ML papers with machine learning models and dataset names and b) tests some baseline models on the annotated dataset. The main contribution is a new data resource.

**Questions For The Authors:**

A. Can you please clarify the input length part for the experiments? You are using paragraphs as units, right?
B. How do you plan to handle corrupt paragraphs in a real-life deployment? Or for that matter, any corruption that comes from pdf to text conversion?

**Reasons To Accept:**

The paper presents a focused contribution that is well explained. I do think the dataset is going to be interesting for future research. The annotation scheme is well thought and a small-scale multi-annotator experiment is reported. The agreement between the annotators shows the validity of the annotation scheme. Two things I like in particular: a) there is a full-text annotation as opposed to just the abstract; b) the paper selection policy is better than random. Also, I do appreciate the fact that in the experiments, the authors use paragraphs as units but cross-validate over the documents to ensure document-wide generalizability of the models.

**Reasons To Reject:**

I believe these are the key weaknesses of the paper:
1. I do have some issues with the applicability of the work. Scholarly entity extraction is indeed an important problem, but if the entities are just dataset names and model names, the contribution becomes too focused. Moreover, we have previously seen that crowdsourcing is an excellent alternative for this problem (paperswithcode).
2. The authors mention that manual annotation is not a very scalable process here (L90-L91). I completely agree. But there isn't any remedy prescribed for that, except an experiment in section 6.2 for training size. Prior work such as SciREX uses PWC as weak supervision, why is that not used here? Without any serious attempt at making the process scalable, there is no extensibility of this work, which severely restricts its utility.
3. Some methodological problems:
a. What authors consider as stacked annotations is a well-known problem called nested named entity recognition (https://aclanthology.org/D09-1015.pdf ) and multi-label classification is not necessary (L449).
b. The baseline models are BERT based, which handles at most 512 tokens, and I am surprised that paragraphs (L435) can be used as the input unit. It is either a) the paragraphs are uncharacteristically small, or b) they are truncated/broken somewhere.
c. While the cross-validation setup helps to test for inter-document generalization, there is probably not going to be a standard train/dev/test split: this will make it very hard for future work to compare against the proposed methods.
d. Scientific entities are emerging, so the test setup needs to capture a temporal aspect: i..e, the models should be tested with documents that are published later than the training ones (hence possibly have some unseen entities).

**Reproducibility:**

3: Could reproduce the results with some difficulty. The settings of parameters are underspecified or subjectively determined; the training/evaluation data are not widely available.

**Reviewer Confidence:**

4: Quite sure. I tried to check the important points carefully. It's unlikely, though conceivable, that I missed something that should affect my ratings.

---

> ### Author Rebuttal · Authors · 2023-08-29
>
> We would like to thank  you for your effort, your time and your detailed, and constructive comments. We have gone through all the comments in detail and discussed all points among all co-authors. Below you will find our rebuttal for the major issues raised by you.
> Our rebuttal is structured as followed. First, we comment on the major points from the “Reasons to Reject” section of your review.  At the end we answer the question of the “Questions to the authors” section. The enumeration refers to the enumeration of the comments/questions brought up by you.
> ## 1. RE your comment on the low contribution of the paper due to that we focus on ML models and datasets only.
> Indeed, with PwC we saw that crowdsourcing  can work well. But, apart from the dataset itself, one of the key contributions of our paper is the annotation guideline and the data model, which both have a detailed granularity than related works. Both are required for a crowdsourcing task for an outcome of certain quality. However, our annotation experiences show that annotation at that level of granularity is hardly crowdsourcable. Moreover, for testing crowdsourced data one will need a very reliable dataset too.
> ## 2. RE your comment on prior works such as SciREX that uses PWC as weak supervision and why is that not used in our paper
>
> While manual annotation is not scalable it produces a work of high quality. In detail, the weak supervision approach, as was used to generate the SciREX dataset, does not convince us. They pre-annotated the full texts with a model trained on a different dataset (SciERC) that has annotations on abstracts only. The reported number of changed annotations  are not aligned with the state-of-the-art model performance on this dataset (f1: 0.71). The used model for pre-annotating  is introducing a bias which is not discussed or handled appropriately.
> So we decided to introduce a new dataset to the community, that is manually annotated from scratch on full text. This could be useful for future usage in annotation approaches like SciREX, incorporating a mixture of weak supervision, automatic pre-annotation and final curation by manual annotators, which focus on scalability.
> ## 3. RE your comment on methodological issues in our work:
> ### a) Stacked vs. Nested annotation
> Thank you for your hint, we will change terminology throughout the whole paper.
> ### b) Input length of paragraphs longer than 512 token (Compare Qustion A)
> In fact we cut at 512 tokens, but the paragraph lengths is shorter than you expect. A Paragraph is defined simply by the output of grobid text extraction. Statistics about the paragraph length (number of tokens using scibert tokenizer as an example) show, that the resulting paragraphs contain ~4.5 Sentences in mean. The number of tokens per paragraph are: in mean: 109.1, median 88. Only 16! Paragraphs are longer than 512token (scibert tokenizer) which results in 25 out of 54,598 annotated entities, which are not able to be annotated when cut after 512 tokens. This is negligible.
> ### c) 10-fold has no standard train/dev/test split for future comparison
> Because of that reason we share our data in a fixed 10-fold setup. (Compare data in download link). We will point this out in the publication itself.
> ### d) Temporal aspects and generalization on unseen entities.
> This is a good hint. Because of the 10-fold setup we do not see a good solution to face that concern in general. But a separate analysis of the generalization over unseen entities is a valid point for future work.
> ## Question from the authors:
> A.	RE your question  about the input length …
> B.	RE your question on how to handle corrupt paragraphs in real-life deployment?
> For this publication we have no final solution for this issue. It is a valid point to discuss in the Limitations. For future real-life deployment we plan to analyse the impact of annotating corrupt paragraphs and develop automatic detection mechanisms.
> ### Comments on missing references:
> Thank you for the hints. We will add the two references into the paper.
>
>
> Thank you for your Review. We are looking forward to incorporating all your valuable hints.

---

### Official Review · Reviewer_AUcQ · 2023-08-10

**Typos Grammar Style And Presentation Improvements:** 1. It might be helpful to include mor…
**Soundness:** 3

**Excitement:**

3: Ambivalent: It has merits (e.g., it reports state-of-the-art results, the idea is nice), but there are key weaknesses (e.g., it describes incremental work), and it can significantly benefit from another round of revision. However, I won't object to accepting it if my co-reviewers champion it.

**Paper Topic And Main Contributions:**

This work focuses on developing a new dataset for the task of identifying machine learning datasets and models in scholarly literature. Unlike prior datasets, they adopt a more fine-grained set of entity types and annotate a corpus of 100 full-text scientific publications called GSAP-NER. Additionally, they also consider informal mentions (such as “our BERT-based model”) as entities, including type annotations for them and allow nested NER annotations. They benchmark the performance of several pretrained models, including both general and science-specific models, on their dataset. Their experiments reveal that while models achieve reasonable performance on this task, some entity types still show a huge performance gap. Finally, they run additional experiments testing model performance on the same task under various low-data regimes to better understand how much data annotation is needed to develop strong models for this purpose.

**Questions For The Authors:**

1. Why are reference links and URLs annotated?
2. Did you try experimenting with RoBERTa-Base? Ideally it would help to compare performance across models of the same size if the goal is to check that performance is better when pretraining domain is closer to dataset domain.
3. Why is the training set size experiment conducted on SciBERT? If SciBERT and SciDeBERTa-CS are the same size, it might be better to take the latter model for these experiments since it has better performance overall.


**Reasons To Accept:**

1. The proposed dataset, GSAP-NER, could be an interesting and challenging new benchmark for the scholarly information extraction community
2. The authors benchmark the performance of a range of general and science-specific BERT-based language models on their dataset and also evaluate performance on the task under limited data settings, highlighting potential directions for future work on this task


**Reasons To Reject:**

One of the key contributions is the development and annotation of a finer-grained set of entity types, however some of the choices are unclear and lack justification. Specifically:
1. It is a little unclear how the distinction between ML model and ML architecture was defined. Based on the examples of GPT, BERT, etc. being considered models, it seems like this distinction is dependent on atomicity, i.e. any entity that involves a combination of multiple architectural choices is considered a model. But it would help to have this notion clarified in the paper as well as the annotation guidelines.
2. The scope of the ML method category is extremely unclear, which could potentially explain the low agreement on this category. Additionally, it’s not clear why we cannot consider method to be the parent category for all model-related spans, and rename this category to miscellaneous? The distinction between model/architecture and method seems a little artificial.
3. The justification for finer-grained dataset typing (i.e. types for resources like KBs, etc. being unclear) is reasonable, but this problem seems to be present even with the new categorization. Are knowledge bases considered to be under datasource since they’re not static? What would dynamic benchmarks or pre-training corpora (C4/Pile) be classified under?
4. Some prior work on entity extraction from CS papers also identify metrics as entities (as mentioned in the paper), however this dataset does not consider that category - what motivated this decision?
5. Why are task entities considered model-related? It seems like including them under dataset-related would be more suitable (considering that later, the same span being labeled as task and dataset is a common observation)

In addition to schema development, the process by which full-text papers are selected for annotation also seems to have some flaws:
1. Instead of sampling a new set of papers, it might have helped to re-annotate a subset of papers from an existing dataset like SciREX.
2. Currently, popular papers are chosen by looking at the most downloaded models on HuggingFace and selecting papers associated with them. However, that biases the sample towards only modeling papers and disregards papers focused on dataset creation, evaluation testbeds, survey papers and reproducibility studies, all of which could contain information about ML methods and datasets. Instead of this metric, popular papers could have been selected based on the highest number of citations (limited to the past X years for recency, if that was another criterion).

Lastly, the paper has very little discussion of what kinds of cases were ambiguous and hard for annotators to agree on. It also does not discuss interesting phenomena in more detail (e.g., spans that models consistently found hard to annotate), which could spark more future work


**Reproducibility:**

4: Could mostly reproduce the results, but there may be some variation because of sample variance or minor variations in their interpretation of the protocol or method.

**Reviewer Confidence:**

4: Quite sure. I tried to check the important points carefully. It's unlikely, though conceivable, that I missed something that should affect my ratings.

---

> ### Author Rebuttal · Authors · 2023-08-29
>
> We would like to thank  you for your ef fort, your time and your detailed, and constructive comments. We have gone through all the comments in detail and discussed all points among all co-authors. Below you will find our rebuttal for the major issues raised by you.
> Our rebuttal is structured as followed. First, we comment on the major points from the “Reasons to Reject” section of your review.  At the end we answer the question of the “Questions to the authors” section.
>
> ## Schema Development (Reasons to Reject 1-5):
> 1.	RE your comment on „the distinction between ML model and ML architecture”
> We see a model as an executable piece of software that can be applied on a dataset . For instance, the mention "BERT-base" is a typical MLModel.
> On the other hand, there are mentions of models that combine architectural choices, like "CNN-LSTM", which we consider as MLModelGeneric with nested ModelArchitecture mentions. We will clarify this in the paper and the guideline.
> 2.	RE your comment on “ML method category is extremely unclear”
> Thank you for the suggestion. We agree that "Method" is a very broad category, which does not distinguish between ML methods from other methods. We have done this in accordance with other annotation guidelines (SciIE used for SciERC), which defines "Method" as a broader category to annotate methodological statements. Introducing a separate entity class "ML Method" could capture an even more fine-grained picture of methodological mentions, which we could introduce in a future version of GSAP-NER . We will improve the description of the entity type "Method" in the annotation guideline and add more examples.
> 3.	RE your comment on the “justification for finer-grained dataset typing”, esp. Datasource vs. Dataset
> We agree with the example you gave: KBs are also considered as Datasources.
> The key distinction (as stated in our guideline) is that Datasource has unstable reproducibility. We will clarify the Datasource definition in the paper and illustrate it with further examples (KG, dynamic Benchm arks). We will also add further examples to the guideline.
> 4.	RE you comment on the missing entity type "Metric".
> Thank you point this out. We are aware of the missing category metric. For this project Metrics and Measures were not in our focus. We will add the entity type Measurements in future, as Leaderboard extraction is part of our future plans.
> 5.	RE your question whether task entities are considered model-related?
> Thank you for your comment. We agree that a task can be related to a model as well as to a dataset. For simplification, we selected the model related category, but we will clarify the connection to a dataset in the entity type description.
>
> ### RE your comment on the paper selection method.
> Thank you for your suggestion. We paid a lot attention to the paper selection method in the beginning of the project and deliberately made it as described, in particular, to focus on two objectives: popularity (first part of the method) and diversity (second part of the method). For these reasons, we did not choose to re-annotate papers from the SciREX corpus. Another reason is that the authors or SciREX do not clearly define their final selection method in the paper, unfortunately. Furthermore, we think that SciREX is also biased towards publications from PWC.
> However, your assumption, that our selection criteria of “popularity” will lead towards having only modeling papers, does not hold. The focus of the popular publications are new models, datasets or focus on specific tasks.
> We will give a short overview of the selected paper including citation count in the Appendix.
> We believe that our paper selection method broadens the set of annotated publications available to researchers in the research domain. Apart from that, the authors of SciREX  do not clearly define their final selection method in the paper, unfortunately.
>
> ### RE your comment on the missing error analysis of annotators disagreement: Hard Cases
> Thanks for this suggestion. We have executed an analysis during the annotation process. We will add insights on particular hard cases: i.e., MLModel vs. Modelarchitecture in the final version of the paper.
>
> ## Questions for the authors:
> 1.	RE your question “Why are reference links and URLs annotated?“
> Future method development could benefit from those annotations. E.g. Disambiguation or better models to detect MLModel or Dataset entities using thiattributive information.
> 2.	RE your question “Why did you not compared to RoBERTa-BASE?”
> You are right, that we did not argue well on the selection of RoBERTa model for comparison. The authors of SciDeBERTa have allready shown the performance gain of using non in domain continual learning of PLMs before finetuning on downsstream tasks. With comparison to RoBERTa-Large we can show, that the finetuned SciDeBERTa-CS model  even perform better than the 3 times bigger RoBERTa model.
> 3.	RE your question “Why not using SciDeBERTa-CS for size experiment?”
> We take the smallest model SciBERT (109M vs 139M Parameters) for this experiment because of the high number of iterations to perform on the 10-fold setup with 10 different trainset sizes (100 Models to train)
>
> Thank you for your review. We look forward to incorporating all your valuable hints.

---

### Meta-Review · Area_Chair_f98r · 2023-09-22

**Recommendation:** 3

**Metareview:**

Reviewers were enthusiastic about the potential of a new NER resource for scholarly information extraction and appreciated the thoughtful approach to designing fine-grained entity classes for machine learning. Several reviewers questioned the schema of NER tags chosen by the authors, particularly raising questions of distinctions between classes such as ML model and ML architecture or Dataset and Datasource, which may also impact annotator agreement. Reviewers also questioned the efficacy of having a small, human-annotated corpus versus other works that have larger, crowdsourced approaches. Reviewers offered many suggestions for improving the writing by correcting terminology, typographical errors, and improving the clarity of ideas. Ultimately, the limitations of the dataset flagged by the reviewers may limit the impact and adoption of this resource by the community.

---

### Decision · Program_Chairs · 2023-10-07

**Decision:**

Accept-Findings

**Comment:**

Reviewers were enthusiastic about the potential of a new NER resource for scholarly information extraction and appreciated the thoughtful approach to designing fine-grained entity classes for machine learning. Several reviewers questioned the schema of NER tags chosen by the authors, particularly raising questions of distinctions between classes such as ML model and ML architecture or Dataset and Datasource, which may also impact annotator agreement. Reviewers also questioned the efficacy of having a small, human-annotated corpus versus other works that have larger, crowdsourced approaches. Reviewers offered many suggestions for improving the writing by correcting terminology, typographical errors, and improving the clarity of ideas. Ultimately, the limitations of the dataset flagged by the reviewers may limit the impact and adoption of this resource by the community.